# Efficiency and selectivity of $CO_2$ reduction to CO on gold gas diffusion electrodes in acidic media

Mariana C. O. Monteiro [1], Matthew F. Philips[1,2], Klaas Jan P. Schouten [2,3 ✉] & Marc T. M. Koper [1 ✉]

The electrochemical reduction of $CO_2$ to CO is a promising technology for replacing production processes employing fossil fuels. Still, low energy efficiencies hinder the production of CO at commercial scale. $CO_2$ electrolysis has mainly been performed in neutral or alkaline media, but recent fundamental work shows that high selectivities for CO can also be achieved in acidic media. Therefore, we investigate the feasibility of $CO_2$ electrolysis at pH 2–4 at indrustrially relevant conditions, using 10 cm$^2$ gold gas diffusion electrodes. Operating at current densities up to 200 mA cm$^{-2}$, we obtain CO faradaic efficiencies between 80–90% in sulfate electrolyte, with a 30% improvement of the overall process energy efficiency, in comparison with neutral media. Additionally, we find that weakly hydrated cations are crucial for accomplishing high reaction rates and enabling $CO_2$ electrolysis in acidic media. This study represents a step towards the application of acidic electrolyzers for $CO_2$ electroreduction.

---

[1] Leiden Institute of Chemistry, Leiden University, Leiden, The Netherlands. [2] Avantium, Amsterdam, The Netherlands. [3] Van't Hoff Institute for Molecular Sciences, University of Amsterdam, Amsterdam, The Netherlands. ✉email: klaasjanschouten@gmail.com; m.koper@lic.leidenuniv.nl

The finiteness of fossil fuel resources and the attempt to minimize the world´s $CO_2$ footprint drive research towards alternative technologies for the production of fuels and chemicals. An important building block employed in large scale for the production of commodity and specialty chemicals is carbon monoxide (CO)[1]. It can be produced via the electrochemical reduction of $CO_2$ ($CO_2$R), and, provided the electricity comes from zero emission technologies, the carbon cycle can be closed[2]. CO is among the most economically viable products that can be obtained through the electrocatalytic reduction of $CO_2$ as it has a high ratio of molecular weight per electron transferred[3–5]. Generating CO electrochemically can enable a less centralized CO production chain by allowing for on-site CO production on demand and eliminate transportation hazards and costs. Nevertheless, in order to achieve $CO_2$ electrolysis to CO at a commercial scale, three main key figures of merit are desired: high current efficiency and current density, low cell potential, and no dependency of these variables on the electrode size[4].

A known limitation of the electrocatalytic $CO_2$R is the low solubility of $CO_2$ in aqueous media (33 mM at 1 atm and 25 °C) and consequently the poor transport of $CO_2$ to the catalyst surface[6]. This limitation can be overcome with the use of gas diffusion electrodes (GDEs), which are porous structures that contain a gas diffusion layer and a catalyst layer[7]. The electrocatalytic reaction happens at the solid-liquid-gaseous interface and the transport of reactants to the catalyst surface is highly enhanced, which allows for achieving higher reaction rates than in conventional systems. Besides the high reaction rates, in order to achieve high current efficiencies for CO, the competing hydrogen evolution (HER) reaction must be suppressed, either by engineering the catalyst/electrode[7] or the reaction environment (electrolyte)[8,9].

The development of $CO_2$ to CO electrolyzers at relatively large scale (GDE area > 5 cm²) is recent, and in most work reported so far the reaction is performed in neutral to alkaline media. In general, the local concentration of various species (such as $CO_2$, $HCO_3^-$, $CO_3^{2-}$ $OH^-$, and $H^+$) has shown to play a crucial role on the competition between $CO_2$ reduction and hydrogen evolution, through proton[10,11], bicarbonate[12], or water[13] reduction. This is relevant as, for instance, the activity of the water reduction reaction at high overpotentials may limit the window in which high current densities can be achieved while sustaining high CO current efficiencies. Industrially relevant current densities have been accomplished at GDEs with a geometrical area larger than 5 cm², however, often at the expense of relatively high cell potentials[14–18]. An additional disadvantage of using neutral or alkaline media is carbonate formation, leading to significant energy and carbon losses[19].

Because of the high activity of proton reduction at low overpotentials, the current efficiencies found for CO in acidic media are usually low, especially on electrodes with low roughness. In fact, only in a few works in the literature, $CO_2$R has been investigated (at small scale) in acidic media[10,11,20,21]. Recent work from Bondue et al.[10] has indicated that proton reduction can be fully suppressed in acidic media, as long as the rate of CO/$OH^-$ formation from $CO_2$R is high enough to compensate the mass transfer of protons to the electrode surface. Differential electrochemical mass spectrometry (DEMS) was used to quantify the formation of CO/$H_2$ and consumption of $CO_2$. The results show that with a high surface area rough gold electrode (R = 20.3), high $CO_2$ pressures (high local concentration of $CO_2$) allow approaching a faradaic efficiency of nearly 100% for CO in mildly acidic electrolyte. A key conclusion from that work was also that $CO_2$ is reduced primarily by $H_2O$, not by protons, even in acidic media, hence leading to the formation of $OH^-$. However, it remains to be seen if the results from this small-scale idealized DEMS study can be translated to high-surface-area gas diffusion electrode systems. Furthermore, our recent work has shown that tailoring the electrolyte cation identity offers an opportunity to favor the water-mediated $CO_2$ reduction over proton-mediated hydrogen evolution in acidic media[21]. Studies carried out in sulfate electrolytes (pH 3) using a flat gold electrode, showed that cations have no effect on the rate of proton reduction, which is only a function of the (local) proton concentration. Meanwhile, the activity for $CO_2$R to CO increases as a function of the cation identity in the order $Li^+ < Na^+ < K^+ < Cs^+$. In addition, performing $CO_2$R in acidic media presents several advantages to a large scale process in terms, for example, of the electrolyte conductivity, $HCO_3^-$ crossover, anode reaction kinetics (in case of oxygen evolution) and electrolyzer design.

In this work we have assessed the feasibility of acidic $CO_2$ electrolysis with practical electrode geometries. Using a 10 cm² gold gas diffusion electrode, we performed $CO_2$R in sulfate electrolytes at different current densities (10–200 mA cm⁻²). Our results show that high CO selectivity (~90%) can be achieved at 100–200 mA cm⁻² in mildly acidic media, as long as weakly hydrated cations are present in the electrolyte ($Cs^+$, $K^+$). Through a direct comparison with neutral media (1 M $KHCO_3$) we find that in 1 M $Cs_2SO_4$ electrolyte, $CO_2$R can be carried out at considerably lower cell potentials, leading to a reduction of up to 30% in process energy costs. Our work represents an important step towards acidic $CO_2$ electrolysis at industrial conditions, with improved energy efficiency.

## Results

$CO_2$ reduction ($CO_2$R) was carried out on gold gas diffusion electrodes, whose schematic structure is depicted in Fig. 1. At the gold-based cathode, $CO_2$ is fed through the back of the electrode and reduced to CO ($CO_2 + H_2O + 2e^- \rightarrow CO + 2OH^-$) at a three phase boundary between the catalyst, the catholyte and $CO_{2(g)}$. At the DSA® anode, water is oxidized to $O_2$ ($H_2O \rightarrow \frac{1}{2}O_2 + 2H^+ + 2e^-$). Depending on the reaction conditions, not all the $CO_2$ supplied to the cathode is converted to CO. Hydrogen ($H_2$) can competitively form either through proton reduction or water reduction, depending on the acidity of the local reaction environment. The local pH near the cathode surface can change

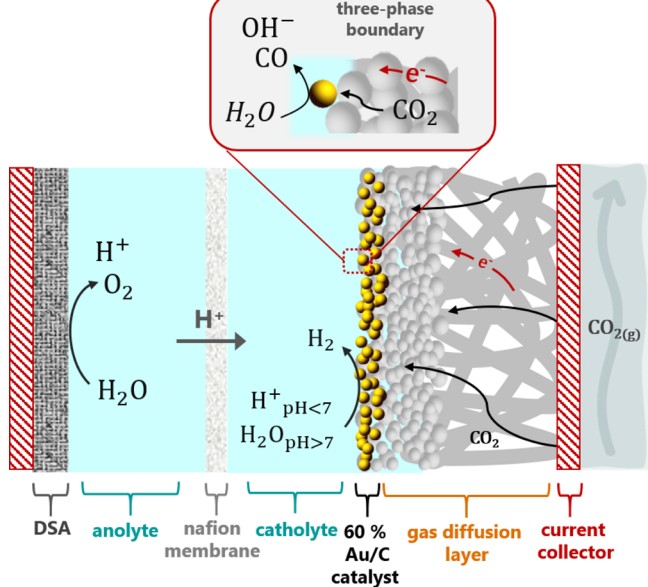

**Fig. 1 $CO_2$ electrolysis.** Schematic representation of the gold gas diffusion electrode system studied in this work.

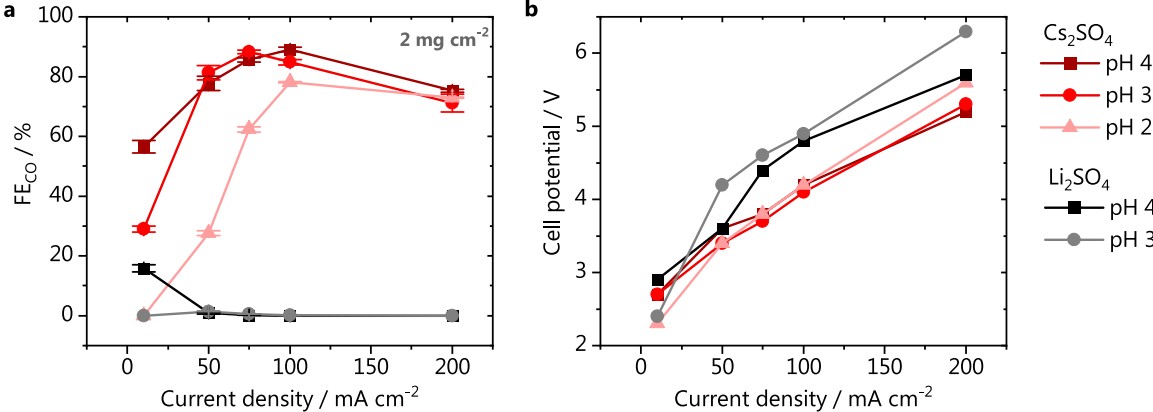

**Fig. 2 Effect of pH and cation identity on CO$_2$ electrolyis. a** Faradaic efficiency for CO and (**b**) cell potential. Electrolysis performed in either 1 M Cs$_2$SO$_4$ (red) or 1 M Li$_2$SO$_4$ (gray scale), catalyst loading 2 mg cm$^{-2}$.

significantly as a function of the current density and reaction selectivity. However, the OH$^-$ produced during CO$_2$ reduction can be neutralized by protons (if the catholyte is acidic) or by the reactant itself, succesively forming bicarbonate and carbonate (CO$_2$ + OH$^- \rightarrow$ HCO$_3^-$; HCO$_3^-$ + OH$^- \rightarrow$ CO$_3^{2-}$ + H$_2$O). Because of these homogeneous reactions, it is important to supply an excess of CO$_2$ in order to achieve high current densities and CO faradaic efficiencies. For all the measurements described in this work, the flow of CO$_2$ (50 mL min$^{-1}$) and electrolyte (30 mL min$^{-1}$) were kept constant, and the anolyte was always 0.5 M H$_2$SO$_4$. The setup did not allow the introduction of a reference electrode, but all system parameters were kept constant in order to allow a comparison of the effect of electrolyte pH on the cathodic reaction. That is: the only parameters varied were on the cathode side, where different electrolytes were investigated and two different loadings of gold nanoparticles were applied to 10 cm$^2$ GDEs, namely 2 and 1 mg cm$^{-2}$.

We have first investigated the feasibility of carrying out CO$_2$ electrolysis on GDEs in acidic media, by performing bulk electrolysis in 1 M Cs$_2$SO$_4$ solutions of pH 2, 3 and 4. The faradaic efficiencies (FE) for CO as a function of the total applied current density can be seen in Fig. 2a. The only products formed were CO and H$_2$, evidenced by the CO and H$_2$ FEs adding up to 100% (Supplementary Fig. 1). Even though formate has been reported as a possible product from CO$_2$ reduction on gold electrodes[22], it was not detected under our working conditions (if it was formed it was below our detection limit). At low current densities, we see that the selectivity for CO is lower at the more acidic electrolyte than at pH 4. However, at 200 mA cm$^{-2}$ >70% FE for CO is obtained, regardless of the bulk pH. A FE of 90% is achieved at pH 4, at 100 mA cm$^{-2}$. In terms of selectivity, these results not only show that large scale CO$_2$R to CO can be accomplished in acidic media, but are also in agreement with the model recently proposed by Bondue et al. based on small-scale DEMS measurements[10]. However, in our case, at low current densities, and low pH, due to the lower mass transport limitation in the GDE configuration, the activity for CO (and corresponding production of OH$^-$) is not high enough to neutralize the protons diffusing towards the catalyst surface, favouring hydrogen production. On the other hand, at current densities above 50 mA cm$^{-2}$, especially for bulk pH 3 or 4, the local acidic environment can be rapidly neutralized[23,24], enabling FEs for CO above 80%. We see that when starting from bulk pH 2, higher current densities are required to achieve this suppression of proton reduction. The slight decrease in selectivity for CO going from 150 to 200 mA cm$^{-2}$ is likely due to the strong alkalinity developing around the catalyst (depletion of CO$_2$ due to the formation of carbonate) and the onset potential of the water reduction reaction. When operating

electrolyzers at high (>100 mA cm$^{-2}$) current densities, low energy costs are crucial to make the process feasible. The cell potentials obtained at pH 2–4 are shown in Fig. 2b, and no significant differences are observed as a function of pH at intermediate current densities, where the selectivities for CO are high. However, at 10 mA cm$^{-2}$, a lower cell potential is found at pH 2, as mainly proton reduction is taking place. The inverse happens at 200 mA cm$^{-2}$, likely due to the lower activity of water reduction. We point out that the measurements at different current densities were performed sequentially, and the GDEs were stable throughout the experiments, i.e., no problems with flooding were encountered.

The cation identity has also been shown to play an important role on the selectivity and activity of CO$_2$ reduction. In neutral and alkaline media, Bhargava et al.[25] have recently reported FEs for CO on silver GDEs increasing from 40 up to nearly 100% depending on the cation used, in the order Na$^+$ < K$^+$ < Cs$^+$ at 50 mA cm$^{-2}$. In acidic media, we have recently shown that the cation identity does not affect the activity for proton reduction, while weakly hydrated cations lead to a drastic increase in the activity for CO[21]. We showed that the cation acts as a necessary promotor for the reaction and that the cation trend for CO formation is related to the higher ability that weakly hydrated cations, such as Cs$^+$, have to coordinate with the adsorbed CO$_2^-$ intermediate, and their high concentration near the electrode surface. In contrast, cations having a strong hydration shell such as Li$^+$, do not strongly coordinate with the oxygen of the adsorbed CO$_2$, and accumulate less near the surface, leading to their inferior promoting effect on the CO production[21]. Therefore, we have also investigated to which extent the electrolyte cation identity plays a role when carrying out CO$_2$R in acidic media in the GDE geometry. The results obtained in 1 M Li$_2$SO$_4$ at pH 3 and 4 are compared to Cs$_2$SO$_4$ in Fig. 2a and the total FEs for CO and H$_2$ are available in Supplementary Fig. 2. Surprisingly, we observe that in Li$_2$SO$_4$ nearly no CO is produced at all applied current densities, regardless of the bulk pH. The low activity for CO$_2$R to CO in Li$_2$SO$_4$ does not allow for reaching high enough CO$_2$R currents at low overpotentials and to hinder the access of protons to the catalytic active sites, yielding nearly 100% FEs for H$_2$ at these current densities. This is confirmed by the higher cell potentials that are observed in nearly the whole current range investigated (Fig. 2b). These findings highlight that differently from neutral and alkaline media, the choice of a cation that favors the production of CO is crucial to enable CO$_2$ electrolysis in acidic media.

We have also prepared a GDE with half the loading of gold nanoparticles than the one used in the experiments shown in Fig. 2. CO$_2$ electrolysis was carried out at the gold GDE with 1

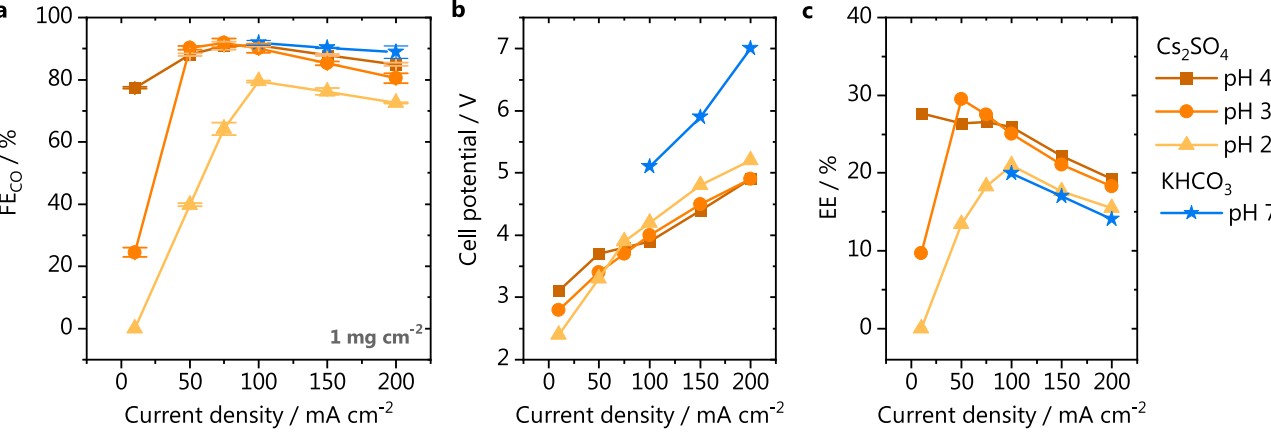

**Fig. 3 Comparison between acidic and neutral media. a** Faradaic efficiency for CO; (**b**) cell potential and (**c**) energy efficiency for $CO_2$ electrolysis performed in either 1 M $Cs_2SO_4$ (orange) or 1 M $KHCO_3$ (blue stars), catalyst loading 1 mg cm$^{-2}$.

mg cm$^{-2}$ loading in the same conditions as before (1 M $Cs_2SO_4$, pH 2–4) and the FEs for CO and cell voltages are displayed in Fig. 3. The total faradaic efficiencies for CO and $H_2$ are shown in Supplementary Fig. 3. Figure 3a shows the same FE trend observed for the 2 mg cm$^{-2}$ GDE as a function of pH. However, the overall values are slightly higher for the lower loading GDE, independent of the bulk pH (see a direct comparison in Supplementary Fig. 4). There are several structural factors of the GDE that could lead to the observed differences. The dispersion of the nanoparticles, for instance, could affect how accessible the catalytic sites are for the build up of the three-phase reaction boundary. Significant differences in thickness of the GDE could also alter the potential drop across the layer and consequently the activity/selectivity. In the literature, there is no consensus about the exact effect that changing the catalyst loading has on the reaction. The work of Bhargava et al.[25] with Ag-based 1 cm$^2$ GDEs suggests that no significant differences are found in terms of CO faradaic efficiency for loadings above 1 mg cm$^{-2}$. In the work of Duarte et al.[17], on the other hand, using 10 cm$^2$ Ag-based GDEs, a higher activity for CO was found in potentiostatic measurements when increasing the catalyst loading from 0.5 to 2 mg cm$^{-2}$. Nevertheless, no significant changes in selectivity were observed. The differences in GDE dimensions and production procedures, in addition to the complex morphologies of these electrodes, currently do not allow drawing a general rule in terms of the effect of the catalyst loading. In view of that, we have analyzed our specific case, by characterizing the morphology and composition of the GDEs used in this work using Scanning Electron Microscopy (SEM) and Energy Dispersive X-Ray Spectroscopy (EDX). The SEM micrographs for the GDEs with 1 and 2 mg cm$^{-2}$ loading can be seen in Fig. 4a and Fig. 4b, respectively. There are no clear differences between the overall morphology of the two GDEs, which can be also seen in the additonal micrographs from Supplementary Fig. 5. However, looking at the dispersion of the gold nanoparticles, it becomes clear that more particle agglomerates are present at the surface of the GDE with the higher catalyst loading. This is further confirmed by elemental analysis (Fig. 4c, d), implying that at the GDE with a lower loading, the nanoparticles are better dispersed on the PTFE matrix. Even though more nanoparticles are present in the GDE with higher loading, these particles are likely less accessible for the reactants, hindering the formation of the three-phase reaction boundary. A schematic representation of these differences is depicted in Fig. 4e, f. It is important to point out that although only one micrograph is shown here, several areas of the GDEs were imaged in order to draw these conclusions. The images

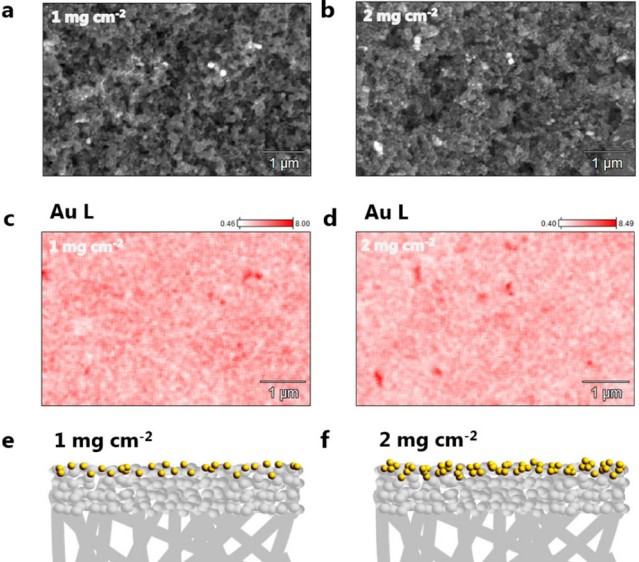

**Fig. 4 Characterization of the GDEs with different catalyst loadings. a**, **b** SEM micrographs, (**c**, **d**) EDX elemental map and (**e**, **f**) schematic representation of the gold GDEs with 1 mg cm$^{-2}$ and 2 mg cm$^{-2}$ loading, respectively.

shown here are representative of the trend found throughout the whole surface. The complete EDX elemental analysis of the GDEs is shown in Supplementary Figs. 6 and 7. The two GDEs have nearly the same composition, in which carbon and fluorine are present in large amounts, due to the PTFE-based gas diffusion layer. In addition, as the analysis was performed after the catalysts were used for $CO_2$ electrolysis (for obtaining a more realistic picture of the GDE during/after operation) cesium and potassium hydroxide deposits are also found homogeneouly distributed in the GDE structure, due to the extreme alkalinity developed at high current densities.

The performance of the 1 mg cm$^{-2}$ GDE in acidic media was compared with its performance in neutral media, by carrying out the electrolysis in 1 M $KHCO_3$, a commonly used electrolyte for $CO_2$R. The total FEs for CO and $H_2$ in $KHCO_3$ are shown in Supplementary Fig. 8. The FEs for CO are compared in Fig. 3a, where we see that the selectivity in sulfate, especially at pH 3 and 4, is nearly the same as in $KHCO_3$ for current densities between 100–200 mA cm$^{-2}$. However, by comparing the respective cell

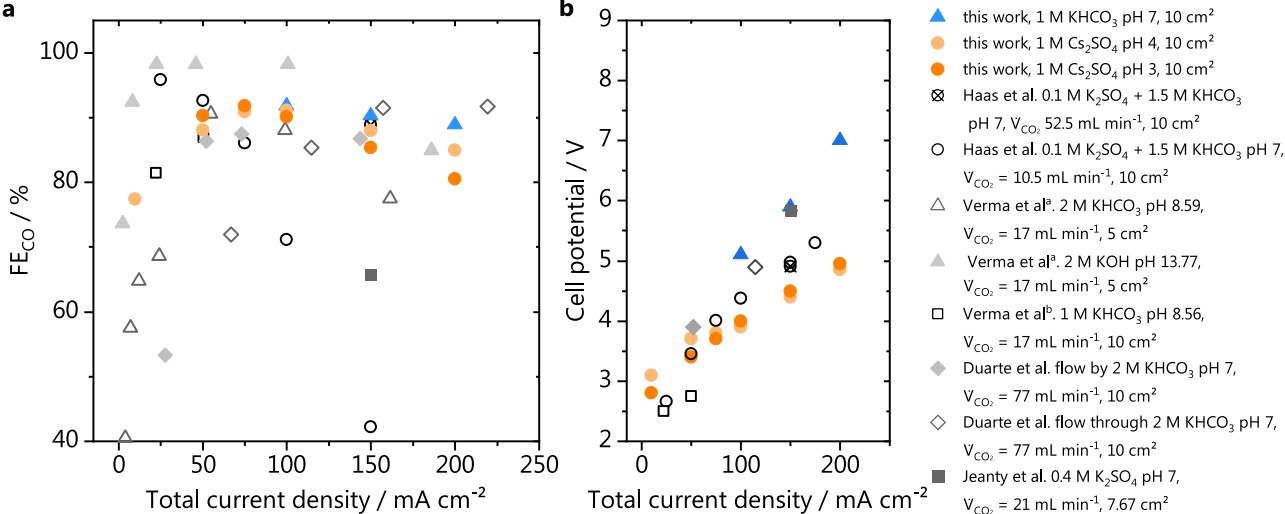

**Fig. 5 Comparison of our results in acidic media with literature values. a** Faradaic efficiencies and (**b**) cell potentials reported in different work from literature, in which $CO_2$ electrolysis was performed in neutral to alkaline media, using GDEs equal or larger than 5 $cm^2$. Results from this work shown in orange circles (acidic media) and blue triangles (neutral media). Data points extracted from the work of Haas et al.[15], Verma et al.[a][13], Verma at al.[b][14], Duarte et al.[16] and Jeanty at al.[17] shown in gray.

potentials (Fig. 3b), we find remarkably lower values in acidic media. This is likely a consequence of the much higher conductivity of the 1 M $Cs_2SO_4$ (234.4 $10^{-4}$ S $m^2$ $mol^{-1}$) electrolyte in comparison to 1 M $KHCO_3$ (117.9 $10^{-4}$ S $m^2$ $mol^{-1}$), which decreases the ohmic losses. Considering that the electricity is the main operating cost of $CO_2$ electrolysis[5], it is crucial that the energy efficiency is optimized in order to make the technology industrially viable. It can be seen in Fig. 3c, that operating the cathode in acidic media, in sulfate electrolyte, leads to a considerable improvement of the process energy efficiency compared to $KHCO_3$ (see Supplementary Equation 1). A similar comparison in terms of the energy consumption per ton of CO produced can be made (Supplementary Fig. 9) and shows that running $CO_2$ electrolysis in $Cs_2SO_4$ at pH 4 instead of $KHCO_3$ (both 1 M) leads to an energy saving of 24, 25 and 30% at 100, 150, and 200 mA $cm^{-2}$, respectively. Considering, for example, an electricity price of 0.03 dollars per kWh[5], running the reaction in $Cs_2SO_4$ at pH 4 instead of $KHCO_3$ (at 200 mA $cm^{-2}$) could lead to saving 1343 dollars per ton of CO produced. The process energetics in sulfate could of course be improved even further, by increasing the ionic strength of the electrolyte in order to reduce ohmic losses, and by refining the cell design.

We have also compared the results of our work, performed in acidic media, with recently reported literature values, for $CO_2$R to CO in neutral/alkaline media performed in gas diffusion electrodes having a geometrical area ≥ 5 $cm^2$. It must be noted that although $CO_2$R is intensively studied, mainly potentiostatic experiments are carried out, which are not representative of an electrocatalytic industrial process. Often, only partial current densities for CO are reported, and no information regarding the cell potential is given. This creates a gap in terms of determining the economical feasibility of the process, and for comparing different reaction conditions and catalysts. Therefore, we have only compared here results obtained in (fairly) similar systems to the one of this work. The main difference is that, in all works reported so far, $CO_2$R to CO has been carried out in catholytes of pH 7 or higher. We have only compared measurements performed at similar scale (5–10 $cm^2$ GDEs) and in electrolytes with moderate to high ionic strength (0.5–2 M). The collection of data points is shown in Fig. 5, for the CO faradaic efficiencies and cell potentials as a function of the total current density. We see in

Fig. 5a that in terms of selectivity, the performance of our GDEs at acidic pH is quite comparable to results obtained in neutral and alkaline media, especially at the higher current densities. On the other hand, in Fig. 5b it can be seen that the cell potentials we obtained in acidic media are considerably lower than the literature values, supporting what we have also concluded based on our own comparison with $KHCO_3$.

## Discussion

Based on our results in acidic media, the system design rules for $CO_2$ electrolyzers, recently presented by Bhagarva et al.[25], should be reconsidered. In their work, pH 14 is pointed out as nearly the sole condition that can yield optimum $CO_2$R to CO performance. However, their comparison is based only on experiments carried out in 1 $cm^2$ GDEs in electrolytes with pH higher than 6. Our work shows that operating in acidic media, specifically in sulfate electrolyte, can considerably decrease the energy costs without harming other important figures of merit such as current density and FE. Table 1 gives an overview of the molar conductivities of different anions and cations that have been used/considered to carry out $CO_2$R. It can be seen that an acidic electrolyte (of a given cation) will always have higher conductivity than an alkaline one. In addition, the less bicarbonate formed during the course of the reaction, the better it is in terms of carbon and efficiency losses. In acidic media, the protons can to some extent neutralize the $OH^-$ produced upon reduction of $CO_2$ while in alkaline media these species will be neutralized by $CO_2$ itself, lowering the reactant´s concentration at the reaction interface. Although pH 14 or larger has been proposed as the optimum condition to carry out $CO_2$ electrolysis[25], it has been recently emphasized that a series of difficulties can arise from operating in hydroxide-based electrolytes at large scale. The work from Nesbit and Smith[26] shows that at high current densities (>1 A $cm^{-2}$) there is a local increase in the electrolyte concentration, which consequently increases its viscosity, slows down diffusion, lowers the electrolyte conductivity (e.g., for KOH concentration >5 M) and $CO_2$ solubility, increases the rate of $HCO_3^-$ production, increases the solute activity and lowers the activity of water. According to the authors, these factors slow down the rate of $CO_2$R and shift the reaction equilibrium potentials. These

**Table 1 Electrolyte compositions for $CO_2$ electrolysis.**

**Conductivity**

| cation | $\Lambda_+$ $10^{-4} \cdot S \cdot m^2 \cdot mol^{-1}$ | anion | $\Lambda-$ $10^{-4} \cdot S \cdot m^2 \cdot mol^{-1}$ |
|---|---|---|---|
| $H^+$ | 349.6 | $OH^-$ | 198 |
| $Rb^+$ | 77.8 | $SO_4^{2-}$ | 160 |
| $Cs^+$ | 77.2 | $CO_3^{2-}$ | 138.6 |
| $K^+$ | 73.4 | $HPO_4^{2-}$ | 114 |
| $Na^+$ | 50.1 | $Br^-$ | 78.1 |
| $Li^+$ | 38.7 | $I^-$ | 76.8 |
| | | $Cl^-$ | 76.3 |
| | | $NO_3^-$ | 71.4 |
| | | $ClO_4^-$ | 67.3 |
| | | $F^-$ | 55.4 |
| | | $HCO_3^-$ | 44.5 |
| | | $H_2PO_4^-$ | 36 |

**Solubility**

| salt | molal* | salt | molal* |
|---|---|---|---|
| KOH | 21.6 | $K_2HPO_4$ | 8.6 |
| $K_2SO_4$ | 6.9 | KBr | 45.0 |
| $K_2CO_3$ | 8.0 | $KNO_3$ | 31.3 |
| $KHCO_3$ | 2.2 | $KClO_4$ | 0.1 |

Equivalent ionic conductivity[29] ($\Lambda°$) of commonly used electrolyte ions, and feasible molality based on the solubility[30] of potassium salts containing different anions.
*Molality (mol kg$^{-1}$) calculated assuming the maximum concentrations feasible according to the solubility values at 20 °C[30].

contributions would be much less significant if operating in an acidic electrolyte. In addition, it is important to point out a few considerations that must be made in terms of the choice of anion for carrying out $CO_2$ electrolysis in acidic media. As shown in Table 1, sulfate anions not only have the highest conductivity (after $OH^-$) but also allow for working in highly concentrated electrolyte. A potassium sulfate solution, for example, can be prepared in concentrations up to 6.8 M. This is not the case for perchlorate, for example, which despite its use in many fundamental electrochemical studies, is poorly soluble and could never be employed in a large scale process. Analyzing Table 1 further, halide anions would be an option considering their high solubility, however, it has been shown that specific adsorption is detrimental for the activity for $CO_2$ reduction. Finally, although hydrogenphosphate has relatively good conductivity and solubility, it has been proposed to act as a proton donor for hydrogen evolution[27].

In summary, here we have shown that, while most of the efforts in the $CO_2R$ field have been on developing better cathode catalysts, the electrolyte design, specifically operating in acidic media, can considerably lower the cell potential and consequently improve the energy efficiency, without compromising the FE for CO. Based on the recent observation in idealized small-scale DEMS measurements that under suitable conditions proton reduction can be suppressed during $CO_2R$, our results show that FEs between 80–90% for CO can be achieved at a gold gas-diffusion electrode at current densities up to 200 mA cm$^{-2}$, demonstrating the feasibility of running $CO_2$ electrolysis in acidic media. Under these industrially relevant conditions, the high rate of $CO_2$ reduction (i.e., high current density) leads to the production of sufficient $OH^-$ to neutralize the protons before they reach the catalyst layer. This hinders proton reduction and enables a high selectivity for CO even at bulk pH 2, at 200 mA cm$^{-2}$. Furthermore, we observe that employing weakly hydrated cations in the electrolyte, such as $Cs^+$ and $K^+$, is crucial for assuring a high $CO_2$ reduction activity in acidic media. Considering operation at industrially relevant scales, there must be a tradeoff between performance and cost, which means $K^+$ sulfate salts are likely more suitable. Finally, by comparing results in acidic and neutral media, we show that considerably lower cell potentials can be achieved at low pH in sulfate electrolyte, which result in energy savings up to 30% operating at 200 mA cm$^{-2}$. Further research is of course necessary in order to lower energy costs even more, achieve higher current densities and fully optimize the process. Our work, however, lays down a new path towards the development of acidic $CO_2$ electrolyzers.

## Methods

**Materials and chemicals**. The following chemicals were used to prepare the electrolytes for bulk $CO_2$ electrolysis: $Cs_2SO_4$ (Sigma Aldrich, 98%), $Li_2SO_4$ (Sigma Aldrich, 98%), $KHCO_3$ (Acros Organics, 99.5%), $H_2SO_4$ (Acros Organics, for analysis ACS, 95% solution in water).

**Gas diffusion electrodes fabrication**. The synthesis method for the gas diffusion layer (GDL) was adjusted from a patented method[28]. First, 15 g of acetylene black (Soltex, 75%-03) was mixed with 8.91 mL PTFE (FuelCellStore, Teflon™ PTFE DISP 30) and 60 mL of a 1:1 volume water/isopropyl alcohol (IPA). After mixing, a dough was obtained and rolled with a marble rolling pin before using a cross rolling technique to obtain the desired thickness. The PTFE dispersion was diluted by 50% with 1:1 volume water/IPA, applied to the back of the dough, and graphitized carbon was placed on top. A Carver heated press (Model number 4533) was used to press the structure at 140 °C and 10.25 Ton for 32.5 min. Then the temperature was raised to 308 °C at the same pressure and held for 32.5 min. Finally the temperature was raised to 317.5 °C at 13 ton and held for 32.5 min before removing the structure from the press. GDLs of 17.16 cm$^2$ were cut from the resulting structure. Commercial gold nanoparticles were used as catalyst (60% gold supported on vulcan XC-72 (carbon), FuelCellStore). The catalyst ink was prepared by suspending the particles in ethanol and adding 250 μL of a Nafion© solution under constant stirring. The inks were additionally sonicated for 30 min. The obtained ink was airbrushed on a 17 cm$^2$ GDL and let dry. The catalysts metal loading was calculated by weighing the electrodes before and after the spraying.

**Gas diffusion electrodes characterization**. The gold GDEs topography and composition were characterized by Scanning Electron Microscopy (SEM) in an Apreo SEM (ThermoFisher Scientific). Micrographs were obtained using an acceleration voltage of 10 kV and an electron beam current of 0.8 nA. Energy Dispersive X-Ray Spectroscopy (EDX) was used for elemental analysis (Oxford Instruments X-MaxN 150 Silicon Drift detector). EDX data processing was done with the Pathfinder™ X-ray Microanalysis software v1.3. The data is displayed in atomic percentage for easier visualization, however, the quantification was performed in automatic mode, without providing external standards.

**Electrolysis measurements**. The gas diffusion electrodes were mounted in a commercial two-compartment 10 cm$^2$ GDE flow cell (ElectroCell, Micro Flow Cell) for the bulk electrolysis. The anolyte and the catholyte were separated by a reinforced Nafion™ membrane N324. For all measurements the anolyte was a 0.5 M $H_2SO_4$ solution and the anode a dimensionally stable anode (DSA®, ElectroCell). The electrolyte flow rate was 30 mL min$^{-1}$, achieved with a peristaltic pump. A $CO_2$ flow of 50 mL min$^{-1}$ was employed for all the measurements. Either 1 M $Cs_2SO_4$, $Li_2SO_4$, or $KHCO_3$ were used as catholyte. The pH of the solutions was adjusted with $H_2SO_4$ when necessary using a pH meter. Galvanostatic bulk electrolysis measurements were controlled using a power supply and each different current density was applied for one hour. The product analysis was performed using a gas chromatograph (Varian 4900 micro GC) equipped with four modules: COx module, MS5 (mol. sieve) module, PPQ (poraplotQ) module and 52 C WAX module. Gaseous samples were taken from the headspace every 3 min. The current efficiencies shown throughout this work represent the average values obtained during 1 h of electrolysis, with the corresponding standard deviation.

## Data availability

Experimental datasets are available from the corresponding author on reasonable request.

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

## Acknowledgements

This work was supported by the European Commission under contract 722614 (Innovative training network Elcorel).

## Author contributions

M.C.O.M., M.F.P. and M.T.M.K. designed the experiments, which were carried out by M.C.O.M. and M.F.P.; M.C.O.M. wrote the paper with input from M.F.P., K.J.P.S. and M.T.M.K. All authors have given approval to the final version of the paper.

## Competing interests

The authors declare no competing interests.
