## [Peer Review File · Nature Communications]

REVIEWER COMMENTS

Reviewer #1 (Remarks to the Author):

In this manuscript, Koper et al. reports selective CO₂ electroreduction to CO in acidic media (pH down to 2) catalyzed by Au gas diffusion electrodes of the size of 10 cm². I think there are particular strengths that justifies publication of this manuscript in Nature Communications. First, effective CO₂ reduction in acidic electrolyte is realized. The overall performance favorably compares to previous works using electrodes of similar sizes and neutral or even alkaline electrolyte. Second, Cs⁺ is found to be important for promoting the CO₂ reduction reaction in this case, which has interesting mechanistic implications. Therefore, I support publication of this manuscript after the following suggestions are considered by the authors.

I note the electrochemical data reported in this manuscript are all from two-electrode systems. I suggest the authors also measure and report the cathode potential. This will help understand the product selectivity, because the competition between CO₂ reduction and HER is clearly a function of the electrode potential. This will also help break down the total voltage of the electrolyzer (CO₂ reduction, OER and internal resistance), which will guide further improvement in device structure and performance.

The authors attribute the selective CO₂ reduction to the less acidic local environment contributed by proton consumption from electrolysis. Is it possible to measure (using spectroscopic tools) or estimate (based on the CO/H₂ ratio) the local pH near the electrode surface?

Reviewer #2 (Remarks to the Author):

In this work, Monteiro and coworkers explore the possibility to design large-scale CO₂ electrolyzers working in acidic conditions. For that, they developed gas diffusion electrodes with gold nanoparticles loaded onto a specific GDL and carried out electrolysis measurements at different pH playing with the supporting salt and the buffer.

While I find the article of interest, it felt much more like an extension of an already existing concept to a more practical dimension. This feeling is re-enforced by the writing in which it is sometimes hard to distinguish the results generated in this work from the literature, and the reader is often left with the impression of reading a review into which few new results from the authors were inserted.

For instance:

The part dedicated to the morphology of the GDE is rather short and non-conclusive. Indeed, outside of observing agglomeration of particles and conclusions such as “particles are likely less accessible for the reactants” to explain the differences seen for different loadings rather qualitative.

Similarly, for the understanding that the performances achieved are similar in acidic conditions than in alkaline conditions regarding current density, while the cell potential is much lower, the fundamental explanation for this observations can be found in a previous paper from the group, lowering the impact

and the novelty of the finding reported in this work.

Overall, if this article was a perspective, or an opinion piece in which the authors would focus on comparing results found in the literature with what they obtained internally (i.e. Figure 5 and Table 1), I would have said that it would be of great interest for the field, alike previous pieces published in this journal (Kanan et al, for instance). Nevertheless, as an original research paper I find the results to lack the insights and the impact required for publishing in a journal such as Nature Communications. I would thus maybe suggest to redraft this article as a perspective, rather than an original scientific paper, and make it even more assertive so to clearly share the fact that alkaline CO₂ electrolysis is presumably not a good choice, and that acidic CO₂ electrolysis appears as a more promising avenue to explore.

Reviewer #3 (Remarks to the Author):

CO₂ reduction studies have been commonly carried out under neutral or alkaline conditions, but acidic conditions have been typically considered to be unfavorable for CO₂ reduction – a view that has only recently started to change. Building on their previous work at a fundamental scale (ref. 10 of the manuscript), the authors investigate here CO₂ reduction at high current densities over gold gas diffusion electrodes using acidic electrolytes. In contrast to the conventional wisdom in the field, they show that high selectivity for CO₂ reduction to CO over a gas-fed cathode is actually possible in an acidic environment as long as the current density is high enough (i.e., >50 mA/cm²), so that OH⁻ ions generated in the reaction neutralize any incoming H⁺ and prevent proton reduction. In agreement with studies in neutral and alkaline media, the authors show that a cation effect on CO₂ reduction also exists in these conditions, with no CO evolution evidenced with a Li₂SO₄ electrolyte. The authors also compare two different catalyst loadings, finding (counter-intuitively) better performance at lower loadings, and carry out experiments with KHCO₃ electrolyte, allowing a direct comparison between acidic and neutral media.

Overall, the manuscript is very well written and the results are presented clearly without any overselling (this is very much appreciated). The experimental methods are described in sufficient detail, and the arguments and conclusions are adequately supported by the presented data. I agree with the authors that there are several potential advantages of operating in an acidic environment, and that the presented results represent an exciting development in electrolyzer and electrolyte design. I think this paper opens new vistas in the field of high current density CO₂ reduction, and that it would be interesting for a broad audience. Consequently, I believe that this manuscript should be suitable for publication in Nature Communications after a few points are addressed. My comments are the following:

1. While I would certainly expect for Li⁺ to show poor results compared to Cs⁺, I find it also very surprising that there is no CO₂ reduction with Li₂SO₄ as the electrolyte. I wonder whether the lithium electrolyte might contain any contaminants that are leading to this result. In our experience, two reagents with the same nominal purity (and from the same manufacturer and “family”) can contain

quite different contaminants. I would recommend the authors to carry out an elemental analysis of the Cs₂SO₄ and Li₂SO₄ electrolytes (for example, by ICP-OES) to verify whether this could be a possible explanation for the poor results with Li⁺.

2. The authors claim that CO and H₂ are the only reduction products formed, as evidenced by the fact that the Faradaic efficiencies add up to 100%. However, there is always some uncertainty in the FE measurements, and previous work in H-cells (at least under neutral conditions) have shown that gold produces small but non-zero amounts of formate (for example, see Cave et al., Phys. Chem. Chem. Phys., 2017, 19, 15856-15863). Are the authors certain that no formic acid was produced, or they did not test the electrolyte for it? (alternatively, it might have not accumulated enough in the electrolyte to reach the limit of detection).

If it is indeed not formed at all, could the authors comment on any possible reasons for this?

3. When comparing the two different loadings, the authors interestingly find that the low-loading electrodes perform better. For example, at 200 mA/cm² the cell voltage with 1 mg/cm² is ca. 4.75 V, while with 2 mg/cm² it is ca. 5.20 V. All other things being equal, this suggests that the 2 mg/cm² cathode has to be polarized to higher overpotentials to reach the same current density. This would occur if, for example, the ECSA of the catalyst at 2 mg/cm² is actually lower than at 1 mg/cm². Do the authors have any measurement of the ECSA of the different electrodes (either by Pb UPD, or double layer capacitance, for example)? This would be very interesting, and it would help confirm their argument about the agglomeration of the catalyst at the higher loading.

4. All experiments are 1-hour electrolyses. Did the authors try any longer tests? How long is it possible to maintain the high selectivity for CO? I am guessing that at some point, flooding will occur and thus only HER is observed. This is not so important in this paper, but since it is also a big issue in the field, any insights on this would be appreciated.

Minor points/corrections:

5. In the manuscript, reference 25 is about a patent for fabricating the GDLs, but the citation is to Hansen's solubility parameters handbook. This is (probably) not correct.

6. Line 295: "Electrolysis measurements" should probably be bold.

7. Using term KPI (which comes from the business literature) is rather awkward. The abbreviation stands for key *performance* indicators (not production, line 63), but overall, it feels somewhat imprecise and out of place. Maybe the authors could use "performance outcomes" (in line 63) or "figures of merit" (in line 229) instead? (the latter is quite standard).

Answers to Reviewer #1 :

In this manuscript, Koper et al. reports selective CO₂ electroreduction to CO in acidic media (pH down to 2) catalyzed by Au gas diffusion electrodes of the size of 10 cm². I think there are particular strengths that justifies publication of this manuscript in Nature Communications. First, effective CO₂ reduction in acidic electrolyte is realized. The overall performance favorably compares to previous works using electrodes of similar sizes and neutral or even alkaline electrolyte. Second, Cs⁺ is found to be important for promoting the CO₂ reduction reaction in this case, which has interesting mechanistic implications. Therefore, I support publication of this manuscript after the following suggestions are considered by the authors.

1.1. I note the electrochemical data reported in this manuscript are all from two-electrode systems. I suggest the authors also measure and report the cathode potential. This will help understand the product selectivity, because the competition between CO₂ reduction and HER is clearly a function of the electrode potential. This will also help break down the total voltage of the electrolyzer (CO₂ reduction, OER and internal resistance), which will guide further improvement in device structure and performance.

In the particular setup used for this study, it was unfortunately not feasible to insert a reference electrode in order to determine the cathode/anode potentials (the same case in large scale electrolyzers). Nevertheless, we assured the main differences in cell potential observed would be due to the process happening on the cathode side. We did that by keeping the cell geometry constant and by using a dimensionally stable anode and a highly conductive anolyte for all measurements. With that, the only variable in the system was the catholyte, allowing us to draw the conclusion that CO₂ reduction to CO can be realized in acidic media, with higher energy efficiency than in bicarbonate electrolyte, which is the main finding of our work. In a recently published fundamental study of our group (<https://doi.org/10.1021/acscatal.1c00272>), one can find more detailed information on the competition between CO₂ reduction and hydrogen evolution on gold electrodes, and how the overpotential, among other variable affects that. This paper is now cited in the manuscript.

We emphasize in the manuscript (lines 65-67) that all system variables were kept constant and only the electrolyte was varied, allowing for attributing the differences observed to what is happening on the cathode side.

1.2. The authors attribute the selective CO₂ reduction to the less acidic local environment contributed by proton consumption from electrolysis. Is it possible to measure (using spectroscopic tools) or estimate (based on the CO/H₂ ratio) the local pH near the electrode surface?

The local pH during CO₂ reduction and other reactions can be measured using different techniques, as scanning electrochemical microscopy (SECM, SIET, SICM), infra-red, rotating

disc electrode, among others. We have recently published a review on the topic (<https://doi.org/10.1016/j.coelec.2020.100649>). We have performed such measurements on flat gold electrodes using Scanning Electrochemical Microscopy (<https://doi.org/10.1021/acs.analchem.9b04952> , <https://doi.org/10.1021/acs.jpcclett.0c02779>) and observed the increase in alkalinity as the CO₂ reduction/hydrogen evolution reactions proceed. Specifically for gas diffusion electrodes, the local pH during CO₂ reduction can only be measured using SECM. Additionally, due to the large roughness of the substrate, a sensitive feedback system must be used in order to control the tip-to-surface distance. This has been achieved and was recently published by Dieckhöfer and co-workers (<https://doi.org/10.1002/chem.202100387>), where the local pH during CO₂ reduction to CO on silver gas diffusion electrodes was measured as a function of potential. They show also an extreme increase in alkalinity, probed only a few nanometers from the surface of the GDEs.

We have now included these references to the sentence mentioned by the reviewer (line 88) in the manuscript, to confirm the increase in alkalinity near the surface during the electrolysis.

Reviewer #2 (Remarks to the Author):

In this work, Monteiro and coworkers explore the possibility to design large-scale CO₂ electrolyzers working in acidic conditions. For that, they developed gas diffusion electrodes with gold nanoparticules loaded onto a specific GDL and carried out electrolysis measurements at different pH playing with the supporting salt and the buffer.

2.1. While I find the article of interest, it felt much more like an extension of an already existing concept to a more practical dimension. This feeling is re-enforced by the writing in which it is sometimes hard to distinguish the results generated in this work from the literature, and the reader is often left with the impression of reading a review into which few new results from the authors were inserted.

We thank the reviewer for his comment. We have indeed illustrated the concept of CO₂ reduction in acid media in a recent paper, but those measurements were under highly idealized model conditions of a flow cell specifically designed for online differential electrochemical mass spectrometry measurements. We consider it an important advance that the conclusions from that work can be extended to a practical gas-diffusion electrode. We would like to note that reviewer 3 actually appreciated our rather factual style (“the results are presented clearly without any overselling (this is very much appreciated).”).

We have taken the reviewer’s comment into account, and in general have highlighted further throughout the manuscript the novelty of our work and how it represents an advance to the field, without trying to oversell.

2.2. The part dedicated to the morphology of the GDE is rather short and non-conclusive. Indeed, outside of observing agglomeration of particles and conclusions such as “particles are likely less accessible for the reactants” to explain the differences seen for different loadings rather qualitative.

As mentioned in the manuscript, in the literature there is no consensus about the exact effect that changing the catalyst loading has on the reaction. Due to the complexity of these gas diffusion electrodes, we have analyzed our own case and based on the SEM images and EDX analysis, which consistently showed more agglomerates in the electrode with higher loading. Therefore we put forward the hypothesis that the catalyst is less accessible to the reactants. The effect of loading was not the focus of this specific work, and the main goal here was specifically to assess the feasibility of large scale CO₂ electrolysis in acidic media. However, we believe the observation we made that a lower catalyst loading leads to slightly improved performance is still important to mention, as it is counter intuitive. We believe that due to the complexity of gas diffusion electrodes, a full understanding of this loading effect requires a separate study.

For the reviewer's information, we have actually recently performed such a separate study on the effect of catalyst loading on the activity of gold gas diffusion electrodes (GDE) for CO. We used Scanning Electrochemical Microscopy (SECM) to map the activity of a gold GDE using a gold nanoelectrode as CO sensor. We approach to only a few nanometers of the Au GDE surface using shear force, and measure the amount of CO that is produced along a catalyst gradient going from low to high loading. The figure below (for review purposes only) shows a preliminary result of this study. Here a line scan was performed for measuring the activity for CO from a low loading (X=0) to a high loading region of the GDE (X=17000). The trends observed point out that increasing the loading has only a small impact in the activity and that other parameters that are responsible for the optimum formation of the reaction three-phase boundary are actually more important. Among these, the accessibility of the catalyst particles shows to be very important, in agreement with our hypothesis in the present manuscript, namely that agglomerates likely hinder the formation of active reaction sites. Another important parameter is the CO₂ back pressure, however, in the present work that was kept constant and could not have played a role. We are currently analyzing the data and preparing a manuscript with this SECM study on the effect of loading.

See changes in lines 115 and 125 of the manuscript.

2.3. Similarly, for the understanding that the performances achieved are similar in acidic conditions than in alkaline conditions regarding current density, while the cell potential is much lower, the fundamental explanation for this observations can be found in a previous paper from the group, lowering the impact and the novelty of the finding reported in this work.

Overall, if this article was a perspective, or an opinion piece in which the authors would focus on comparing results found in the literature with what they obtained internally (i.e. Figure 5 and Table 1), I would have say that it would be of great interest for the field, alike previous pieces published in this journal (Kanan et al, for instance). Nevertheless, as an original research paper I find the results to lack the insights and the impact required for publishing in a journal such as Nature Communications. I would thus maybe suggest to redraft this article as a perspective, rather than an original scientific paper, and make it even more assertive so to clearly share the fact that alkaline CO₂ electrolysis is presumably not a good choice, and that acidic CO₂ electrolysis appears as a more promising avenue to explore.

The other work from our group that reports the faradaic efficiency for CO in acidic electrolyte (<https://doi.org/10.1021/jacs.0c10397>) makes no prediction about the overpotential or cell potential, only about the selectivity. Those results were obtained in a very different flow cell used for DEMS, and cannot directly be translated to a larger scale system. In the DEMS electrochemical cell, large diffusion layer thickness are found under the working conditions, facilitating the depletion of protons at the reaction interface. In the GDE configuration, as we see in our results, much larger current densities are required in order to achieve this effect. Still, in terms of CO faradaic efficiency, we find that results from such an idealized DEMS model study, can be translated to a 10 cm² gas diffusion electrode. However, the DEMS study does not make predictions about the energy efficiency of operating in acidic media. The main advance of the current work is showing that CO₂ electrolysis can be carried out in acidic media, at high current densities, with GDEs. Beyond that, another main result is that we find that the cell potential is significantly lowered by operating in acidic media, due to the superior conductivity of the acid electrolyte. This was concluded not only based on our work, but also when comparing with results from literature (Figure 5). We believe our work creates a new branch in the field and opens up numerous possibilities for optimizing the process further.

We have taken Reviewer's #2 comment into account and highlighted in the main text the difference between our work and what has been previously published in literature (lines 38-40, 51, 72, 77-80, 87-89). We also highlighted the corresponding advance to the field, in agreement also with the positive comments made by Reviewers #1 and #3 on this aspect.

Reviewer #3 (Remarks to the Author):

CO₂ reduction studies have been commonly carried out under neutral or alkaline conditions, but acidic conditions have been typically considered to be unfavorable for CO₂ reduction – a view that has only recently started to change. Building on their previous work at a fundamental scale (ref. 10 of the manuscript), the authors investigate here CO₂ reduction at high current densities over gold gas diffusion electrodes using acidic electrolytes. In contrast to the conventional wisdom in the field, they show that high selectivity for CO₂ reduction to CO over a gas-fed cathode is actually possible in an acidic environment as long as the current density is high

enough (i.e., $>50 \text{ mA/cm}^2$), so that OH^- ions generated in the reaction neutralize any incoming H^+ and prevent proton reduction. In agreement with studies in neutral and alkaline media, the authors show that a cation effect on CO_2 reduction also exists in these conditions, with no CO evolution evidenced with a Li_2SO_4 electrolyte. The authors also compare two different catalyst loadings, finding (counter-intuitively) better performance at lower loadings, and carry out experiments with KHCO_3 electrolyte, allowing a direct comparison between acidic and neutral media.

Overall, the manuscript is very well written and the results are presented clearly without any overselling (this is very much appreciated). The experimental methods are described in sufficient detail, and the arguments and conclusions are adequately supported by the presented data. I agree with the authors that there are several potential advantages of operating in an acidic environment, and that the presented results represent an exciting development in electrolyzer and electrolyte design. I think this paper opens new vistas in the field of high current density CO_2 reduction, and that it would be interesting for a broad audience. Consequently, I believe that this manuscript should be suitable for publication in Nature Communications after a few points are addressed. My comments are the following:

3.1. While I would certainly expect for Li^+ to show poor results compared to Cs^+ , I find it also very surprising that there is no CO_2 reduction with Li_2SO_4 as the electrolyte. I wonder whether the lithium electrolyte might contain any contaminants that are leading to this result. In our experience, two reagents with the same nominal purity (and from the same manufacturer and “family”) can contain quite different contaminants. I would recommend the authors to carry out an elemental analysis of the Cs_2SO_4 and Li_2SO_4 electrolytes (for example, by ICP-OES) to verify whether this could be a possible explanation for the poor results with Li^+ .

We rule out the lower activity for CO in Li_2SO_4 comes from contaminants, as in the EDX analysis performed after the electrolysis, no foreign metal deposition was observed on the GDE (e.g. common contaminants as Fe, Pb, Ni, Al). We believe this lack in activity for CO in Li_2SO_4 on the GDE configuration, can be explained based on our other work (under revision) specifically on the effect of cations on the competition between CO_2 reduction and proton reduction (Reference 21). We see that cations do not have any effect on proton reduction (as expected), while they are necessary for the CO_2 reduction reaction to CO to happen. The cation trend for CO comes from the higher ability of weakly hydrated cations as Cs^+ have to coordinate with the adsorbed CO_2^- intermediate, and their high concentration near the electrode surface. In contrast, cations having a strong hydration shell as Li^+ , nearly do not coordinate with the oxygen of the adsorbed CO_2 , and accumulate less near the surface, leading to the inferior activity for CO . The work of Bondü (Reference 10) shows that high faradaic efficiencies for CO can only be achieved if the rate of CO_2 reduction is sufficiently high (which is not the case in Li^+ containing electrolytes).

We explained the results regarding the cation effect better, in lines 97-101, 106-107, based on Reference 21 (our paper under revision), to make clearer why the selectivity for CO is so poor in Li_2SO_4 .

3.2. The authors claim that CO and H_2 are the only reduction products formed, as evidenced by

the fact that the Faradaic efficiencies add up to 100%. However, there is always some uncertainty in the FE measurements, and previous work in H-cells (at least under neutral conditions) have shown that gold produces small but non-zero amounts of formate (for example, see Cave et al., Phys. Chem. Chem. Phys, 2017, 19, 15856-15863). Are the authors certain that no formic acid was produced, or they did not test the electrolyte for it? (alternatively, it might have not accumulated enough in the electrolyte to reach the limit of detection).

If it is indeed not formed at all, could the authors comment on any possible reasons for this?

Indeed there are previous works on gold where non-zero amounts of formate were produced. However, under our working conditions formate was not detected. Indeed as the reviewer mentioned it might not have accumulated enough in the electrolyte to reach detectable amounts and if it was then formed, it was in negligible amounts.

We have now added this statement to the main text, so we do not rule out entirely the possibility of the formation of traces of formate (lines 75-77).

3.3. When comparing the two different loadings, the authors interestingly find that the low-loading electrodes perform better. For example, at 200 mA/cm² the cell voltage with 1 mg/cm² is ca. 4.75 V, while with 2 mg/cm² it is ca. 5.20 V. All other things being equal, this suggests that the 2 mg/cm² cathode has to be polarized to higher overpotentials to reach the same current density. This would occur if, for example, the ECSA of the catalyst at 2 mg/cm² is actually lower than at 1 mg/cm². Do the authors have any measurement of the ECSA of the different electrodes (either by Pb UPD, or double layer capacitance, for example)? This would be very interesting, and it would help confirm their argument about the agglomeration of the catalyst at the higher loading.

Indeed the knowledge of the electrochemically active surface area could help further understanding the differences found for the two catalyst loadings. However, for these gas diffusion electrodes it is not feasible to accurately determine the ECSA, as due to the porous nature of the GDE, and hydrophobic nature of the PTFE matrix, it is difficult to predict if all the gold nanoparticles applied to the GDE are wetted when the GDE is in contact with the electrolyte. Also it is difficult to predict if the wetting is the same when e.g. doing Pb UPD and when the GDE is being polarized to very cathodic potentials and at high current densities. Therefore, for this manuscript, we drew our hypothesis only based on the SEM/EDX analysis, which gave clear insights. In another study, we have recently used Scanning Electrochemical Microscopy (SECM) to map the activity of a gold GDE as a function of catalyst loading, using a gold nanoelectrode as CO sensor. We have prepared a GDE substrate containing a well-known/controlled gradient of the same Au nanoparticles used in the present work. We approach to only a few nanometers of the Au GDE surface using shear-force, and measure the amount of CO that is produced along this catalyst gradient going from low to high loading. Preliminary results of this study already point out that increasing the loading only has a small impact in the CO activity and that other parameters that are responsible for the optimum formation of the reaction three-phase boundary are actually more important. Among these, the accessibility of the catalyst particles shows to be very important, in agreement with our hypothesis in the present manuscript, that agglomerates likely hinder the formation of active reaction sites. Another

important parameter is the CO₂ back pressure, however, in the present work that was kept constant and could not have played a role. We are currently analyzing the data and preparing a manuscript with this SECM study specifically discussing the effect of loading and other parameters on the local CO activity of GDEs. Please refer also to answer 2.2 to Reviewer 2.

3.4. All experiments are 1-hour electrolyses. Did the authors try any longer tests? How long is it possible to maintain the high selectivity for CO? I am guessing that at some point, flooding will occur and thus only HER is observed. This is not so important in this paper, but since it is also a big issue in the field, any insights on this would be appreciated.

From our own experiences using our homemade substrates, flooding is much more related to the gas diffusion layer morphology and composition, than to the electrolyte pH. Therefore, for this specific work, we have not performed longer tests, as mentioned by the reviewer, it was not relevant for this manuscript. Here we were mainly focused on the feasibility of the reaction at the 10 cm² scale in acidic media. Still, even though we measured only 1 hour at each current density, the different current densities were applied subsequently, which means that each GDE could withstand more than 8 hours of operation without flooding occurring. In fact, within the time scale of our measurements, flooding was never a problem, due to the high stability of the gas diffusion layer used (Patent Reference 30: Philips, M. F., Davide, A., Figueiredo, M. C. & Krasovic, J. Method for the preparation of a gas diffusion layer and a gas diffusion layer obtained or obtainable by such method. 24 (2020).).

Comment added on this point on lines 115-117.

Minor points/corrections:

3.5. In the manuscript, reference 25 is about a patent for fabricating the GDLs, but the citation is to Hansen's solubility parameters handbook. This is (probably) not correct.

We thank the reviewer for this observation. We have now included the correct citation.

3.6. Line 295: "Electrolysis measurements" should probably be bold.

Corrected, line 256.

3.7. Using term KPI (which comes from the business literature) is rather awkward. The abbreviation stands for key *performance* indicators (not production, line 63), but overall, it feels somewhat imprecise and out of place. Maybe the authors could use "performance outcomes" (in line 63) or "figures of merit" (in line 229) instead? (the latter is quite standard).

Corrected, changed to "figures of merit" in line 9 and line 190.

REVIEWERS' COMMENTS

Reviewer #2 (Remarks to the Author):

In their rebuttal, the authors have extensively discussed the novelty of this work. I do believe that indeed switching scales and demonstrating the feasibility of running CO₂ electrolysis in a, industrial format is a strong message the deserves publications in Nature Communications. However, I still believe that it would have been better to gather all the information on one single paper rather than having multiple papers in parallel.

Anyway, I am glad to provide my support for publication of this work as I am certain that it will bring very important discussions in the field.

Reviewer #3 (Remarks to the Author):

The authors have satisfactorily addressed all my inquiries and comments. I commend the authors for their interesting study and a well written manuscript, and I look forward to seeing it in Nature Communications.

Answers to the reviewers (in red)

Reviewer #2 (Remarks to the Author):

In their rebuttal, the authors have extensively discussed the novelty of this work. I do believe that indeed switching scales and demonstrating the feasibility of running CO₂ electrolysis in a, industrial format is a strong message that deserves publications in Nature Communications. However, I still believe that it would have been better to gather all the information on one single paper rather than having multiple papers in parallel. Anyway, I am glad to provide my support for publication of this work as I am certain that it will bring very important discussions in the field.

We thank the reviewer for pointing out the novelty of the work and recommending the manuscript for publication.

Reviewer #3 (Remarks to the Author):

The authors have satisfactorily addressed all my inquiries and comments. I commend the authors for their interesting study and a well written manuscript, and I look forward to seeing it in Nature Communications.

We thank the reviewers for the recommendation.